# Deformation Mechanism of Depositing Amorphous Cu-Ta Alloy Film via Nanoindentation Test

**DOI:** 10.3390/nano12061022

**Published:** 2022-03-21

**Authors:** Weibing Li, Xiao Wang, Xiaobin Feng, Yao Du, Xu Zhang, Yong Xie, Xiaoming Chen, Yang Lu, Weidong Wang

**Affiliations:** 1ZNDY of Ministerial Key Laboratory, Nanjing University of Science and Technology, Nanjing 210094, China; njustlwb@163.com; 2School of Mechano-Electronic Engineering, Xidian University, Xi’an 710071, China; wangxiao_9626@stu.xidian.edu.cn; 3CityU-Xidian Joint Laboratory of Micro/Nano-Manufacturing, Shenzhen 518057, China; xbfeng2-c@my.cityu.edu.hk; 4Department of Mechanical Engineering, City University of Hong Kong, Kowloon 999077, Hong Kong SAR, China; 5Hubei Key Laboratory of Theory and Application of Advanced Materials Mechanics, Wuhan University of Technology, Wuhan 430070, China; 6Department of Materials Science and Engineering, Northwestern University, Evanston, IL 60208, USA; yaodu2015@u.northwestern.edu; 7Department of Mechanical Engineering, Northwestern University, Evanston, IL 60208, USA; xuzhang2017@u.northwestern.edu; 8School of Advanced Materials and Nanotechnology, Xidian University, Xi’an 710071, China; yxie@xidian.edu.cn; 9Micro- and Nanotechnology Research Center, State Key Laboratory for Manufacturing Systems Engineering, Xi’an Jiaotong University, Xi’an 710049, China; xiaomingchen@xjtu.edu.cn

**Keywords:** Cu-Ta alloy, deform mechanism, magnetron sputtering, nanoindentation, molecular dynamics (MD)

## Abstract

As a representative of immiscible alloy systems, the Cu-Ta system was the research topic because of its potential application in industry, military and defense fields. In this study, an amorphous Cu-Ta alloy film was manufactured through magnetron sputter deposition, which was characterized by scanning electron microscopy (SEM) and transmission electron microscopy (TEM). Mechanical properties of Cu-Ta film were detected by the nanoindentation method, which show that the elastic modulus of Cu3.5Ta96.5 is 156.7 GPa, and the hardness is 14.4 GPa. The nanoindentation process was also simulated by molecular dynamic simulation to indicate the deformation mechanism during the load-unload stage. The simulation results show that the structure <0,2,8,4> and <0,2,8,5> Voronoi cells decreased by 0.1% at 50 Ps and then remained at this value during the nanoindentation process. In addition, the number of dislocations vary rapidly with the depth between indenter and surface. Based on the experimental and simulation results, the Voronoi structural changes and dislocation motions are the key reasons for the crystallization of amorphous alloys when loads are applied.

## 1. Introduction

An important player in the alloy family, metallic glasses (MG) were first prepared by Duwenz et al. [1] in 1960. They obtained the first amorphous alloy chips in the Au-Si system by means of the liquid spray technique. Subsequently, Cohen et al. [2,3,4] proposed criteria for the formation of metallic glasses and verified the presence of glass transport in amorphous alloys in terms of specific heat theory. Since then, more amorphous alloys based on different metals have been prepared [5,6,7,8]. In addition, these amorphous alloy systems have excellent mechanical strength and high elastic deformation.

Copper-based amorphous alloys, a branch of MG, have excellent strength and elasticity capabilities. Much attention has been paid by many researchers to its preparation and mechanical characteristics. As early as 1989, Subrananlan et al. [9] introduced the phase diagram of the Cu-Ta binary alloy, elucidating that Ta is difficult to miscible in Cu. Several preparation methods have been used to produce Cu-Ta alloys, including ball milling, ion beam mixing, physical vapor deposition and magnetron sputtering [10,11,12,13,14]. Depending on the concentration of Ta, Cu-Ta systems can form crystal clusters or amorphous alloys. Ta films can act as diffusion barriers for Cu-based semiconductor substrates [15,16,17]. Francis et al. [18] investigated the growth of Ta films in (011) body-centered cubic (BCC) Ta and the deposition of CuTa films on (111) face-centered cubic (FCC) Cu. Frolov et al. [19] investigated the effect of Ta concentration on grain growth and mechanical strength in nanocrystalline (NC) Cu-Ta alloys. Li et al. [20] evaluated the axial tensile strength and elastic modulus of NC Cu-Ta alloys by molecular dynamics methods, and from their results it appears that the strength of the alloy varies with the grain size in the Hall-Petch effect. In particular, many researchers have given more attention to the interface between the amorphous state and NC. Gu et al. [21] prepared CuTa/Cu multilayers with different nanoscale Cu layer thicknesses and investigated the structure-related size effects by nanoindentation tests. Capenter et al. [22] prepared CuNi multilayer films by magnetron sputtering and in which the deformation mechanisms of the CLS (Constrained Layer Slip) model and grain boundary excitation were explained. According to their conclusions, increasing the concentration of nickel contributes to the contribution of CLS and slip to hardness.

Meanwhile, nanoindentation plays a key role in nanoscale mechanisms as an important characterization method. Ruestes et al. [23] used three different interatomic potentials for indentation simulations of tantalum disulfide and investigated the effects of indentation tip and penetration velocity by molecular dynamics (MD) methods. Zeng et al. [24] examined by nanoindentation methods Ta content of 25.6 to 96 at% of the mechanical properties of Cu-Ta amorphous films. They found that the incremental elastic modulus per Ta atom was 1.35 GPa and the incremental hardness per Ta atom was 0.205 GPa. Fang et al. [25] investigated the deformation behavior of high entropy bodies by nanoindentation MD simulations. In addition, many researchers have studied the mechanical properties of different materials by nanoindentation experiments and MD methods [26,27,28,29,30,31,32,33,34,35]. 

Although the mechanical properties at the interface between MG and NC in Cu-Ta alloys have been investigated in the above studies, the details of the deformation have not been precisely explained. In the present study, amorphous Cu-Ta alloy films were first prepared by magnetron sputtering. The deposition and nanoindentation processes were then simulated by MD methods. It was found that the Voronoi cell has a strong influence on the amorphization of the metal. Finally, the deformation behavior and the motivation of the interfacial structure are discussed.

## 2. Experimental

### 2.1. Magnetron Sputter Deposition

Cu-Ta films were synthesized by direct current (DC) magnetron sputter-deposition in an argon atmosphere (99.99%) using Ta (99.5% purity) and Cu (99.5% purity) targets. We used 1.8 cm × 1.8 cm silicon substrates. The substrates were ultrasonically cleaned in acetone and methanol bath for 10 min sequentially. The base pressure of the deposition chamber was about 1.3 × 10^−5^ Pa. Deposition was performed at an Ar pressure of 0.4 Pa, substrate bias of −100 V with the substrate held at ambient temperature. The target power was set at 100 W for Ta and 50 W for Cu, respectively. The Cu concentration in the film was controlled by adjusting the duty cycle of the shutter in front of the Cu target. The thickness of all films was about 1 μm, as determined by optical system.

### 2.2. Nanoindentation

Cu-Ta alloy films were indented using Hysitron 950 Triboindenter with a Berkovich tip (about 50 nm radius). The samples were cut to fit the microscope stubs and combine them using Ag paste. 2 × 3 array was selected on each sample. The maximum depth was about 100 nm less than 10% of films thickness to avoid substrate influence. The load resolution of the instrument is 30 nN and the displacement resolution is less than 0.2 nm. After the test, the area function was corrected on a standard sample of fused silicon.

### 2.3. Microscopy

A FEI QUANTA 450 SEM was used in ultrahigh resolution mode to investigate surface morphology and element distribution. The SEM was operated at 10 kV in high vacuum mode and the sample concentrations were detected by energy-dispersive spectroscopy (EDS). The lamellae were produced with a FEI Helios 600i dual-beam FIB/SEM, operated at 30 kV of Ga+ source and 10 kV of electron source. A JEM-2100 F TEM operating at 100 kV acceleration voltage was used to determine orientation relationship and microstructure of the amorphous Cu-Ta alloy film. As the Figure 1 shows, there are fewer particles on the surface of the Ta-rich film. Figure 1c shows more clearly that the thickness of the Cu-Ta alloy film is 1.087 μm.

### 2.4. Theroy and Modeling

#### 2.4.1. Elastic Modulus and Hardness

We can obtain the elastic modulus and hardness from the indentation test. The most popular method calculating modulus is O&P method [36]. This method fits the unload curve of indentation range from the bottom of unload points to 40% of unload curve. Details are shown below.
(1)$$P_{u}=a{\left(h-h_{r}\right)}^{m}$$
where *a* is the fitting parameter, *h*_r_ is the residual depth, and the index *m* is the shape parameter of the indenter. Stiffness *S* can be determined by derivative load *P*_u_ as Equation (2) shows:(2)$$S=\frac{dP_{u}}{dh}{|}_{h=h_{\max}}=ma{\left(h_{\max}-h_{r}\right)}^{m-1}$$

The reduced elastic modulus and true elastic modulus can be obtained from the next 2 equations, and *A* represent the contact area.
(3)$$E_{r}=\frac{\sqrt{\pi }}{2}\cdot \frac{S}{\sqrt{A}}$$
(4)$$\frac{1}{E_{r}}=\frac{1-v_{s}^{2}}{E_{s}}+\frac{1-v_{i}^{2}}{E_{i}}$$
where *E*_r_ is reduced elastic modulus, *E*_s_ and *E*_i_ represent elastic modulus of material and indenter, respectively. *v*_s_ and *v*_i_ are passion ratio of material and indenter. It is worth mentioning that some other authors [37,38] recommend using a model for all penetration depth/thickness ratio. These models are mostly based on weight function, depending on coating and substrate modulus. Thus, composite modulus is close to both initial moduli.

The O&P method is very popular in the indentation experiment results. However, in the atomic scale, this method may be failure [39]. In MD simulations of nanoindentation, most researchers derive the modulus of elasticity and nanohardness from Hertz theory [40]. According to Hertz, the starting point of the load-depth curve is the absolute elastic phase. In this phase, the load and depth satisfy the following relationship. It should be noted that the derivation of the following equations is only for molecular dynamics simulations of nanoindentation experiments in this work.
(5)$$P_{s}=\frac{4}{3}E^{*}R^{\frac{1}{2}}h_{t}{}^{\frac{3}{2}}$$

Here, *R* is the radius of virtual sphere indenter, *h_t_* is the total indent depth. *E** is the reduced elastic modulus (To distinguish from the real experimental reduced modulus *E*_r_). Since in our model the radius of indenter can’t far exceed the indent depth, according to Pathak et al. [41,42] the last equation should be modified as bellow:(6)$$P_{s}=\frac{4}{3}E^{*}\frac{{\left(Rh_{t}-h_{t}^{2}/4\right)}^{\frac{3}{2}}}{R}$$

According to the Equation (5), the reduced elastic modulus can be obtained by fitting the curve range from the start to the first dislocation emission.

Again, as mentioned above, the load-depth curve can be converted into a strain-stress curve according to Hertz theory. Details are given below.
(7)$$\sigma =\frac{P}{\pi \left(Rh_{t}-h_{t}^{2}/4\right)}$$
(8)$$\varepsilon =\frac{{\left(Rh_{t}-h_{t}^{2}/4\right)}^{\frac{1}{2}}}{R}\cdot \frac{4}{3\pi }$$

From the two equations above, we can easily obtain the elastic modulus in the linear phase.

#### 2.4.2. Simulation Model

In order to explore the deformation mechanism of Cu-Ta alloy film, the simulated sputter depositions and nanoindentation were executed by the Large-scale Atomistic/Molecular Massively Parallel Simulation (LAMMPS) [43]. The embedded atom model (EAM) developed by Foiles [44] and Ravelo [45] are used to describe the force on Cu-Cu pair and Ta-Ta pair, respectively. The angular-dependent interatomic potential (ADP) [46] was set to describe the function of Cu-Ta pair.

As with the sputter deposition experiments, a copper substrate was set up at the bottom of the system. The substrate is divided into three zones, including a fixed zone, a thermally controlled zone and a free zone. In the thermally controlled area a Langevin thermostat is set to maintain the temperature at 300 K. The injected atoms leave the incidence plane and reach the free zone. Two important sections are set up at the top of the simulation box. The incident layer is located one lattice distance from the entire top of the box. As shown in Figure 2, two virtual walls are set up to prevent the emitted atoms from leaving the system. The frequency of the emitted atoms was controlled so that the Ta content approximately met that of the experimentally prepared amorphous film. The boundaries are set to be periodic in the x and y directions and contracted in the z direction.

Next, the output of the deposition simulation can be used as the initial model for the indentation. The indentation model is shown in Figure 3. A virtual sphere indenter with a radius of 3 nm is set at the top center of the deposition output model. The distance between the indenter edge and the top of the deposition output model is 1 Å. The speed of the indenter downwards is set to 2 Å/s. At the start of the simulation, energy minimization and relaxation were performed to bring the system to an equilibrium state with minimum energy. During the indentation test, the model boundary was set to be periodic in the X and Y directions and contracted in the Z direction.

## 3. Results and Discussion

In the results and discussion, it is divided into two parts: experiments and simulations. The experimental part shows the structure of our prepared amorphous Ta-rich Cu-Ta films and the results of nanoindentation experiments. In addition, the focus is on the analysis of the results of the atomic simulations to obtain the deformation mechanism of the deposited films during the indentation process.

### 3.1. Experimental Results

The internal structure of the prepared Cu3.5Ta96.5 films was characterized using a JEM-2100 F TEM. First, a 1 µm slice was cut out at the interface between the film and the substrate using Focused Ion beam (FIB), and the thickness of both the substrate and the film was about 500 nm. As can be seen in Figure 4a,b, there is no apparent array arrangement inside the film, which is in a distinctly amorphous state. To further verify this, the red circle in (a) was characterized by selected area electron diffraction and the image was found to be a distinct aperture as shown in Figure 4d, which further confirms that the prepared film is amorphous. Figure 4c shows the EDS spectrum of the atomic composition of the film, confirming the Ta content.

The results of the nanoindentation experiments are shown in Figure 5, which can be illustrated by the complete load-displacement curve. The hardness and elastic modulus are 14.4 GPa and 156.7 GPa, respectively.

### 3.2. Atomic Simulation Results

In the atomic simulation results, we discuss the results of the simulated deposition and analyze the changes of the Voronoi cell during the deposition process. In addition, we simulated the nanoindentation process of the deposited film, and as a complement to the experimental results dynamically analyzed the motion behavior of the internal atoms.

#### 3.2.1. Deposition Properties

In order to observe the activity of atoms and the changes in internal structure during deposition simulation, we used the Voronoi index and radial distribution function (RDF) to characterize the changes in the crystal structure of the model during deposition. At the same time, we characterized the internal structure and crystal properties of the prepared Cu-Ta films by High Resolution Transmission Electron Microscope (HRTEM). As shown in Figure 4, it can be seen that the Cu3.5Ta96.5 films obtained by sputtering have an amorphous structure. This is also well supported by the results of the atomic simulations. Figure 6 shows the statistics of the change in the number of the first five Voronoi structures in the whole model at different sputtering times. It can be seen that the proportion of <0,12,0,0> cells keep decreasing as the deposited atoms are continuously injected. This is because the <0,12,0,0> units correspond to FCC structures centered on copper atoms. Before deposition starts, single-crystal copper is used as a substrate and <0,0,12,0> also occupies the largest proportion. In addition, <0,5,4,0> is a common defect structure to be observed in single crystal copper. At the end of deposition, the proportion of <0,12,0,0> reaches a minimum of 6.3%. In addition, the proportion of <0,6,0,8> centered on tantalum atoms increases from 0 to 55.8% with the implantation of atoms, making it the largest Voronoi unit in the whole model. It was shown that the degree of amorphization during metal vitrification is associated with Voronoi polyhedral with a particular structure [48]. In our simulations, the variation of <0,6,0,8> polyhedral clearly affects the vitrification process of Cu-Ta alloy films.

To demonstrate the amorphous structure of the films from macroscopic statistics of the atomic arrangement. We use RDF to describe the atomic distribution inside the film after different deposition times. Combined with the distribution of Voronoi cells at the corresponding time in Figure 7, the atomic composition inside the film at this time can be analyzed. Figure 7d shows the atomic distribution at 1800 Ps deposition. The smaller probability of finding a Cu atom at the same distance through the RDF of the Cu-Cu pair corresponds to the smallest proportion of Voronoi cells centered on Cu atoms. g(r) fluctuations between Cu-Ta and Ta-Ta are smaller and the amorphous character is more pronounced.

#### 3.2.2. Indentation Properties

The elastic modulus and nanohardness in the atomic simulations were obtained by Hertz theory. Figure 8a shows the load-displacement curve for the entire indentation process with a maximum indentation load of 0.51 μN. the maximum indentation depth is 2.8 nm. the residual depth is 2 nm. a short distance at the beginning of the loading phase is linearly increasing, indicating that the film is in the fully elastic deformation phase. The elastic modulus can be obtained by fitting a curve for this stage. Figure 8c shows the results of the Hertz fit for this phase, with a reduced elastic modulus of 111 GPa. In addition, by using Equations (8) and (9), the load-displacement curve can be converted to a stress-strain curve as show in Figure 8d, and then the curve for the linear phase can be fitted to obtain the reduced elastic modulus, which gives a result of 109 GPa, in agreement with (c). However, it differs from the experimental value (156.7 GPa) by 30%. After analysis, this difference may be caused by the amplification of the “substrate effect”. During the indentation process, the simulated nanoscale film system has only a very small thickness (8 nm). If we use h (depth of indentation)/t (film thickness) as the normalized evaluation criterion, the experimental value is less than 10%, while the simulated value reaches 30%. The “substrate effect” at the atomic level would have a more pronounced effect. Therefore, the modulus values obtained by scaling up the simulated system will be accurate. Optimizing the simulated deposition times will be closer to the experimental values. When the strain is less than 0.1, the film is in the fully elastic deformation phase, where the strain also corresponds to the initial emission position of the dislocation. Figure 8b shows the load variation corresponding to different loading times. The three sub-images correspond to the depth of the indentation at different times and are color coding with indentation depth.

The nanohardness is calculated using the following formula.
(9)$$H=\frac{P}{A}$$
where *P* is the max load, *A* is project area.

The nanohardness of the Cu3.5Ta96.5 film obtained was 12.6 GPa, a difference of 12% from the experimental value of 14.4 GPa. The H/E (hardness-to-elastic modulus ratio) is a key indicator of the film’s wear resistance. The lower the value, the lower the abrasion resistance and the higher the wear resistance. The simulated Cu-Ta alloy film had an H/E of 0.114, while the prepared film had an H/E of 0.092. The results of the simulated system were also relatively close to the experimental value, which could reflect the high wear resistance of the Ta-rich film.

#### 3.2.3. Deformation Mechanism

Figure 9 shows the change in depth of the film in the Z-direction during dynamic loading and unloading. The color of the atoms is determined by the depth in the indentation direction. The indentation depth increases with increasing load and reaches a maximum at an indentation depth of 2.8 nm. The corresponding picture is shown in (e). The force unloading process is then carried out. As the load decreases, the film bounces back. However, due to the effect of plastic deformation, the indentation depth no longer decreases but remains at this value when the indentation depth is 2.0 nm. The indentation depth at this point is also the residual indentation depth *h*_r_.

To show more clearly the displacement of atoms inside the film during the indentation process. Figure 10 shows the displacement of atoms in the XY-plane for different indentation depths using the slice method. The color of the atoms is determined by the size of the displacement. Figure 10c corresponds to a maximum indentation depth of 2.8 nm and (d) to the state of the atoms inside the model after unloading. At the same time, there is a clear “indentation” during loading, as shown by the red outline in Figure 10c. As the indenter penetrates into the film, the surrounding surface of the film bulges, forming a small convex surface. This is the “Pile-up” phenomenon, where it is known that the Oliver-pharr method is no longer applicable, which is why we use the Hertz model in simulation. The distribution of atoms during the indentation process shows that the atoms on the original free surface have two trends of movement: one is the part of the atoms that moves downward with the indenter, i.e., the red part, and the other is the green atoms that occur in the Pile-up region. The green atoms are bulged by the joint action of the indenter and the surrounding atoms to form a convex package. After the unloading is completed, it can be seen from Figure 10d that some of the red atoms diffuse into green or yellow atoms due to the elastic effect. There are also light blue atoms in the lower part of the entire hemisphere due to plastic deformation, and according to their distribution, it can be approximated that if the indentation load continues to increase, these areas will become sprouts for crack expansion.

Figure 11 shows the displacement of the atoms during the indentation process, we then used the dislocation analysis (DXA) method [49] to calculate the dislocations in the XY plane at a distance of 1.5 nm from the indentation surface. At the same time, the variation of the Voronoi cells was accounted for. In general, the variation of the Voronoi cells during the indentation process is not very significant and most of the structures remain at a stable value. Only the Voronoi cells of <0,2,8,4> and <0,2,8,5> decreased by 0.1% at 50 Ps and then remained at this value. The change in the number of dislocations allows a better analysis of the changes in the internal structure of the film. It can be seen that there are still many defects in the initial state model. As the indenter penetrates deeper into the substrate, the number of in-plane dislocations changes significantly. From 80 Ps onwards, the number of dislocations increases significantly, which also corresponds to the end of the fitted curve in Figure 8c. In our calculations, we consider this moment as the time of initial dislocation emission. Subsequently, the number of dislocations at 160 Ps decreases rapidly. At this moment, the indenter starts to leave the substrate surface and the internal dislocations decrease. However, a few dislocations still occur during the recovery of the film. The number of dislocations remains constant until the indenter leaves the substrate completely.

## 4. Conclusions

Firstly, amorphous Cu-Ta alloy films, namely Cu3.5Ta96.5, were prepared by magnetron sputtering. The surface morphology and internal atomic structure of the films were characterized by SEM and TEM. The TEM results showed that the prepared Cu-Ta alloy films were amorphous, but some nano-clusters were present in them. The elastic modulus and hardness of the film were obtained by nanoindentation. the hardness and modulus of Cu3.5Ta96.5 were 14.4 GPa and 156.7 GPa, respectively. it is evident from the indentation experimental results that the elastic modulus and hardness increase with increasing Ta concentration. Then, both atomic deposition and nanoindentation were simulated using the MD method. The results of the atomic deposition simulations were compared with the TEM characterization results to further demonstrate the amorphous structure of the films. By analyzing the variation of Voronoi units, the main structure affecting the amorphization of the Cu-Ta system is probably the <0,6,0,8> structure centered on the Ta atom. The elastic modulus and nanohardness of the simulated nanoindentation were calculated using a Hertz fitting method. The results obtained were compared with the experimental results and the numerical differences were found to be within 30%. This may be due to the “substrate effect” of the single crystal copper substrate during the simulation and the size of the simulated system. Finally, the specific deformation behavior of the Cu-Ta alloy films was shown by analyzing the changes in atomic displacements during nanoindentation and the number of dislocations in the XY plane. The results show that the Voronoi unit and the number of dislocations per unit area dominates the deformation behavior of the material together.

## Figures and Tables

**Figure 1 nanomaterials-12-01022-f001:**
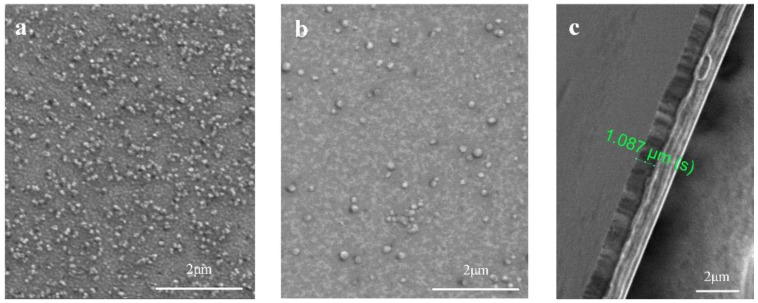
Surface morphology of Cu-Ta alloy films. (**a**) The surface of Cu84Ta16 (where 84 represents the atom concentration of Cu is 84%); (**b**) The surface of Cu3.5Ta96.5; (**c**) The thickness of Cu-Ta alloy film (where the sample is Cu84Ta16).

**Figure 2 nanomaterials-12-01022-f002:**
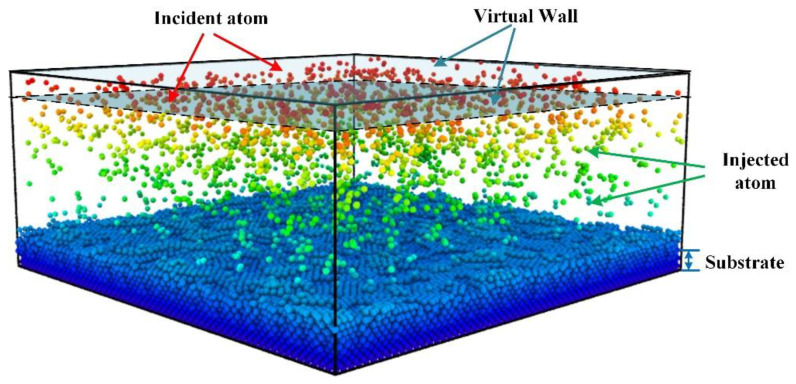
Physical model of sputter deposition. The bottom blue atoms represent Cu atoms. The region red atoms located is the birthplace of incident atom. The atoms in the middle of the simulation box are the emitted atoms before they reach the free surface.

**Figure 3 nanomaterials-12-01022-f003:**
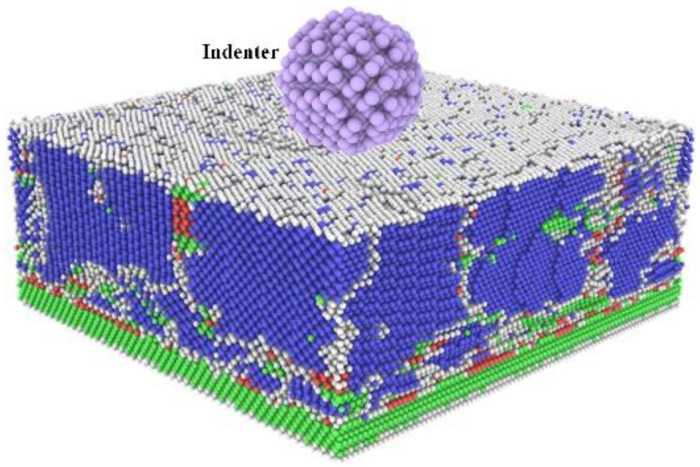
Physical model of nanoindentation. The top purple sphere represents the virtual indenter. The bottom block is output model of sputter deposition, which color coding with common neighbor analysis (CNA) (Reproduced with permission from [47]. Journal of Physical Chemistry, 1987). The green and blue atoms represent FCC and BCC structure type, respectively.

**Figure 4 nanomaterials-12-01022-f004:**
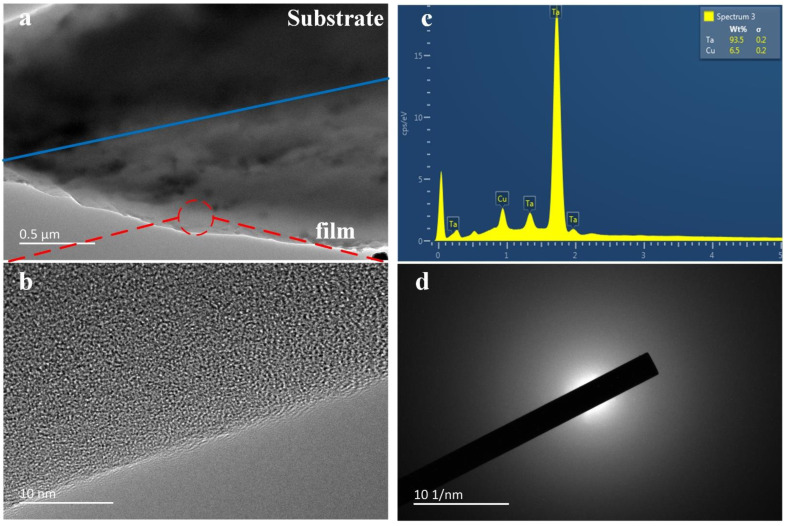
TEM characterization of Cu3.5Ta96.5 film. (**a**) The inner structure of Cu-Ta alloy film; (**b**) A partial enlargement of red circle in (**a**), obviously amorphous state; (**c**) The EDS spectrum of Cu3.5Ta96.5; (**d**) Selected area diffraction patterns sampling Cu3.5Ta96.5.

**Figure 5 nanomaterials-12-01022-f005:**
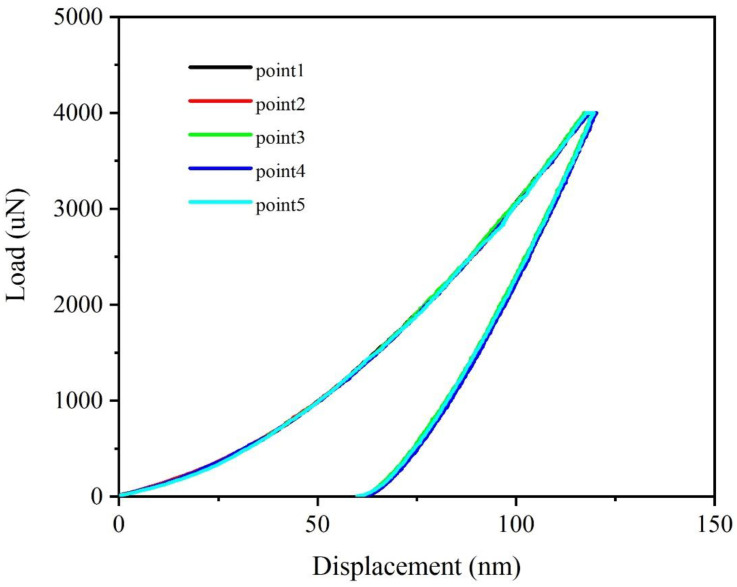
Load-displacement curves of nanoindentation experiment for Ta-rich film. Five points in sample were detected and the elastic modulus and hardness were obtained from these curves.

**Figure 6 nanomaterials-12-01022-f006:**
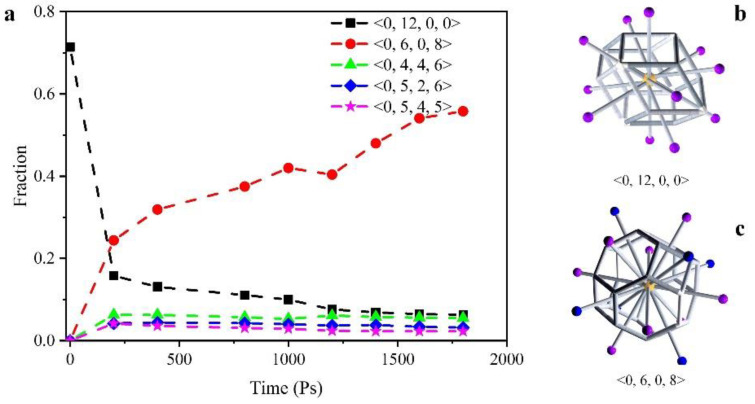
The change of Voronoi polyhedron with the increase of deposited atoms. (**a**) <0,12,0,0>, <0,6,0,8>, <0,4,4,6>, <0,5,2,6> and <0,5,4,5> fractions with the deposit time; (**b**) the structure of <0,12,0,0> Voronoi cell; (**c**) the structure of <0,6,0,8> Voronoi cell. (For structural clarity, the central atom is indicated in yellow and the other atoms in purple or blue).

**Figure 7 nanomaterials-12-01022-f007:**
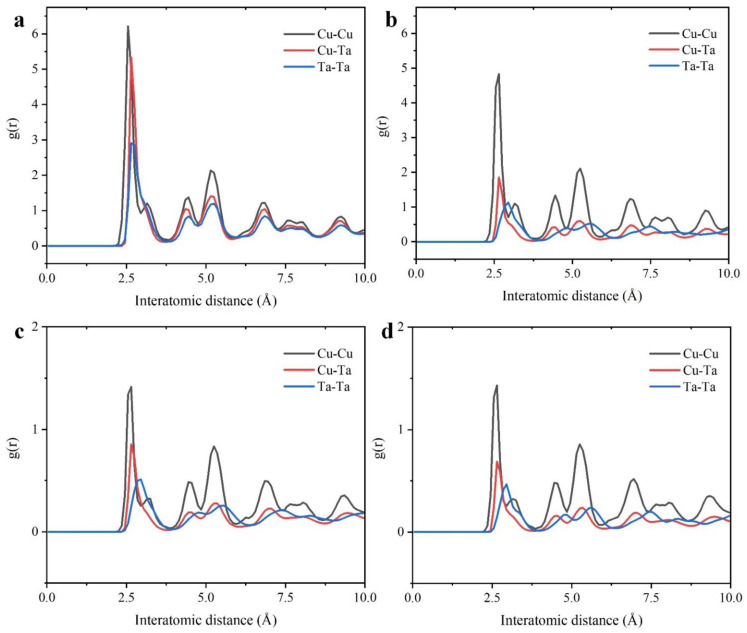
The RDF profile of deposit Cu-Ta model at different time. (**a**) 200 Ps; (**b**) 800 Ps; (**c**) 1200 Ps; (**d**) 1800 Ps.

**Figure 8 nanomaterials-12-01022-f008:**
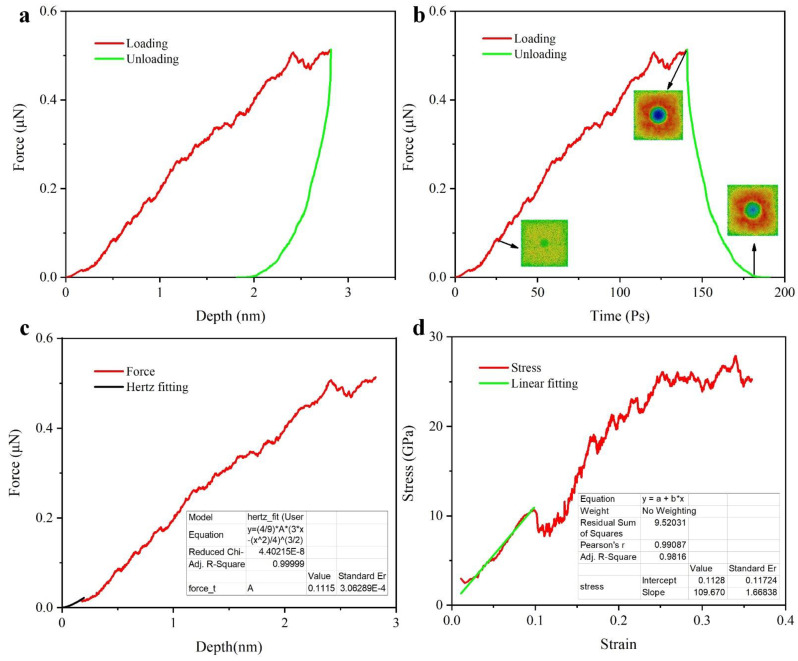
Indentation load curve. (**a**) Load-depth curve of simulated Ta-rich deposition output model; (**b**) Time-force of nanoindentation, three sub-images relate to the indentation depth at that time; (**c**) Load-depth curve of loading stage and Hertz fitting; (**d**) Strain-stress curve transfer from load-depth curve.

**Figure 9 nanomaterials-12-01022-f009:**
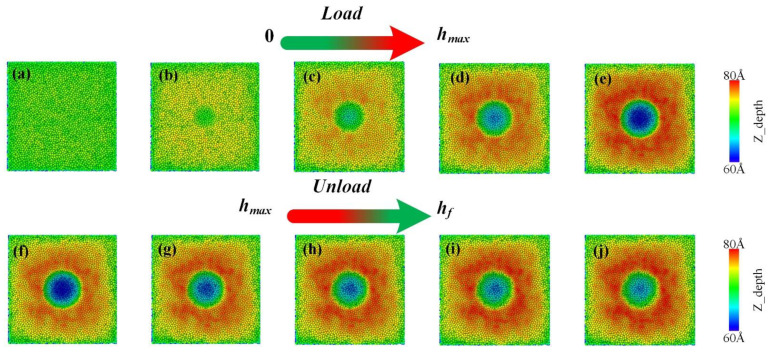
Dynamic loading and unloading process. (**a**–**e**) represent the process of load increase, (**f**–**j**) represent the process of unloading. The atoms are colored according to the depth in the loading direction (Z direction).

**Figure 10 nanomaterials-12-01022-f010:**
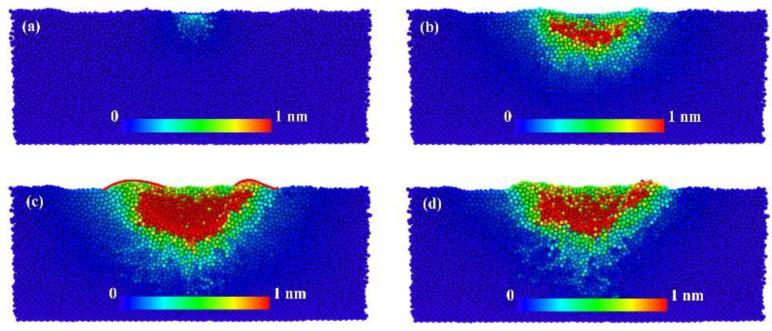
The atomic displacement of the Cu-Ta alloy film during nanoindentation process are the cross-section snapshots of XY plane at the indentation depths of (**a**) 0.2, (**b**) 1.2, (**c**) 2.8 and (**d**) 2 nm, respectively.

**Figure 11 nanomaterials-12-01022-f011:**
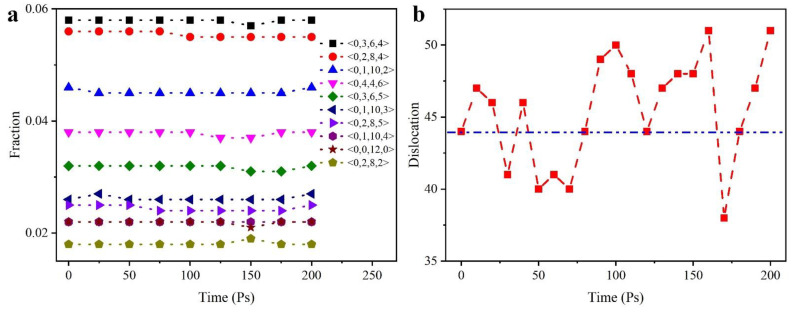
Changes in the internal structure of the model during indentation. (**a**) Voronoi cell statistics at different indent time; (**b**) Dislocation count in XY plane at different indent time.

## Data Availability

Not applicable.

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
