# Peer review of "Deformation Mechanism of Depositing Amorphous Cu-Ta Alloy Film via Nanoindentation Test"

_nanomaterials, 2022, doi:10.3390/nano12061022_

Round 1

Reviewer 1 Report

This paper deals with mechanical behavior of a Cu-Ta alloy, investigated by means of indentation, experiments and simulations. The paper is quite difficult to read, experimental and numerical parameters are difficult to distinguish. Experiments are conducted with a Berkovich tip, but simulation are performed with a sphere. It is not clear if Berkovich tip is considerd a sphere for shallow penetration. Definition of hardness is not clear and it seems to be calculated in elastic regime. Concerning the experimental part some severe drawbacks need to be corrected or clarified before publication.

  • Few errors are reported here:

l42 : « They obtained the first […]”

capital letters are missing (i.e. l. 55 “Ta” and more)

“where 84 represents the concentration of Cu is 84%”, this sentence is not very clear

  • Some scientific comments are presented in the following.

2.2. Nanoindentation:

Five indents are a lower limit for repeatability in nanoindentation, usually a batch of ten indents is performed. Why the array of indent take the shape of a cross (“four points of a square and the center point of it”)?

The maximum penetration is set below 10% of the film thickness. This (empirical) rule fits for hardness measurement, but not for Young’s modulus where a limit of 1% is usually admitted. Therefore, values of Young’s modulus given in the following are affected by the properties of the substrate, and consequently are not corresponding to the coating. It exists several multilayer model that could take into account for the coating and substrate contributions.

If maximum penetration is limited to 100 nm, that means diagonals are approximately 700 nm. What is the resolution of the optical system to be able to accurately measure such a small distance?

2.4.1 Elastic modulus and hardness

l134: A is the projected contact area? Er is the reduced elastic modulus? equivalent to E* (equations 5, 6)?

l140: a reference could be added for the Hertz model.

l146: what is ht in the equation 6? how is obtained equation 6? the Hertz’s equation (5) is well known but it is not clear how authors can re-write it into equation 6.

What is the value of R? Is the Berkovich tip considered as a sphere for small penetration depth? This radius should be characterized.

3.2. Indentation properties

Authors use the theory of Hertz to determine the hardness. Hertz model is an elastic model whereas hardness is linked to plastic activities. There is here something a bit illogical.  

The first part of simulated indentation curves (figure 8) are fitted with an hertzian behavior. According to the figure 8c, it seems that the fit is performed on approximately 1Angstrom! What physical meaning could be given to such a small penetration?!

“The simulated nanoscale films had a more pronounced effect on the substrate during the indentation process than the experimentally prepared micron-scale thickness.” The meaning of this sentence remains unclear. The coating is deposited on silicon substrate, with an approximately 200 GPa Young’s modulus. Since both experimental (156 GPa) and simulated (109 GPa) Young’s modulus of coating are below 200 GPa, it is probably in the case of the experiment that substrate has a stronger influence. This may be linked to the high penetration depth, as discussed above.

In equation 9, what is A? projected area?

3.3. Deformation mechanism

Authors describe a “shrink-in” phenomenon around indentation contact. But, according to figure 10 (especially c and d) it more likely to be a “pile-up” behavior. Thus Oliver and Pharr model, based on a “sink-in” behavior, is not the right one to use to process load-displacement curves.

Reviewer 2 Report

The authors present the study of amorphous Cu-Ta film by nanoindentation test. The paper provides some interesting results but it has to be significantly improved before publication:

The authors compare experimental nanoindentation with Berkovich indenter and numerical simulation with spherical indenter. The tip of the Berkovich indenter is for sure rounded, but authors must provide information about this rounding (diameter or area). Without this, the comparison with numerical simulations is not justified. Also experiments are done up to 100 nm while simulations only to 2 nm.

References for equations 5-8 must be provided.

The authors try to describe the evolution of Voronoi cells during indentation. However, there is no comparison with experimental data. 

English has to be improved. 

For example: (line 144) "Because in our model the radius of indenter can’t far more than the indent depth, the last equation should be modified as bellow." There is a missing verb in the sentence.

"simulant" is not a proper word

"point" - location in Fig.5

Reviewer 3 Report

Authors represented a report of both experiemntal and theoretical investigations of magnetron sputtered Cu-Ta composite film. The manuscript is well written with exceptions of some minor errors, typos; some of those are listed below but I advice authors to re-read text once more. However, the manuscript is structured and prepared good. The presence of both experiment and modeling for such investigations are a great advantage. In my opinion, though this article is very narrow and area-specific, it represents new data that should be published.

1) Line 83-85: the sentence should be rephrased as it can also claim authors were the first one to prepare magnetron sputtered Cu-Ta films.

2) In abstract authors claim that Voronoi cell changes plays role in crystallization, but in conclusions states that this has little effect.

3) There are some small typo errors in text, I advice Authods to carefully check the text again.

4) Can you please describe why the chosen ration between Cu and Ta in the composite film were chosen? Was there a specific behaviour? Were there higher number of samples but you decide to demosntrate only two with the most significant results?

Reviewer 4 Report

Experimental patterns and developed numerical models of the processes of measuring indentation and sclerometry of the "coating-substrate" system can be useful in studying and predicting the processes of deformation and destruction of various materials, as well as studying their mechanical characteristics. Therefore, the article has scientific and practical value.
At the same time, there are several questions and recommendations to improve it:
1. The captions on the figures are very small. They are very difficult to read and analyze. I suggest stretching the pictures to fit the width of the page. Rules for the design of articles allow this.
2. It is known that the properties of the substrate material directly affect the nature of the elastoplastic deformation and fracture of the film, as well as its adhesive (critical force during coating delamination) and cohesive strength (critical force during failure of the surface layer of the film). It is necessary to describe these patterns in more detail in the article.
3. It is shown that the indentation diagrams (Fig. 8) contain jumps corresponding to the fracture of the material and delamination of the film from the substrate. It is necessary to describe in more detail the mechanism of delamination of the coating, the presence of cracks, etc.
4. The regularities of indentation shown in fig. 10 are described insufficiently, which complicates the understanding of the processes of localization of deformation and failure of solids analyzed in the article.
5. 124 …. hardness
6. The article often uses the name of the property of the material "hardness", while "nanohardness" is being investigated. It is known that the scale level of hardness must be specified, since this parameter may differ at different scale levels. I propose to study this issue on the basis of an article  https://link.springer.com/article/10.1007/s11015-013-9680-6, and if this article is useful to you, add its analysis to the Introduction.

Round 2

Reviewer 1 Report

Authors answered to comments, but there is still some parts to be imporved. From a general point of view, the link between experimental and simulation is hard to see whereas both protocol are somehow confusing.
Here is some new comments to the answers of authors:

Q2 : Some others authors (i.e. International Journal of Solids and Structures Volume 44, Issues 25–26, 15 December 2007, Pages 8313-8334) recommend to use a model for all penetration depth / thickness ratio. These models are mostly based on weight function, depending on coating and substrate modulus. Thus, composite modulus is therefore close to both initial modulus. It could be presented like this and the reference of Ranjan Saha added to the manuscript too.

Q3: Thus the Berkovich tip is calibrated following the procedure of Oliver and Pharr, by means of indentation in a reference material? It seems the radius value of 50 nm is given from a general point of view and not as an “actual” value in this experiment (see answer to Q7). It would be clearer for the reader to rewrite this part and remove the reference to an equivalent sphere at the apex of Berkovich if it is not used in the following.

Q4: Ok that’s clearer now, it could be interesting to write it like this in the paper.

Q7: Ok, for me it is not enough clear in the text the “frontier” between experimental and simulation. Maybe by adding a sentence it could become clearer.

Q10: I understood that authors compare penetration depth / thickness ratio. And they conclude that ratio is higher in the case of simulation than in experiment. But how can they explain that Young’s modulus is smaller in the case of simulation. Considering that Young’s modulus of substrate is greater than both simulated and experimental values. If “substrate effect” is higher for the simulation, the corresponding modulus shoud be closer to the one of substrate.

Q12: The Hertz model doesn’t take into account for pile-up (at least in elastic regime). What is about the experimental? Do pile-up have been observed?

Author Response

Please see the attachment。

Reviewer 2 Report

The authors provided some new information in the revised manuscript. However, the connection between the experimental and simulation is still not justified.

The limitations of simulation cell size are fully understandable and the simulations are correct. But the experiments are performed on the order of magnitude higher length scale and the results cannot be simply compared with each other. Materials exhibit a strong size effect during nanoindentation, especially for small depths. Therefore I suggest adding experimental results of indentation with different maximal depths to see, how the hardness and modulus are changing. Or the other option is to remove these experiments from the paper and focus solely on the simulation part.

reference (37) is in Chinese - the authors should provide a reference in English.

Reviewer 4 Report

Accept.

Author Response

Thanks for your valuable comments!